# PLUG-AND-PLAY GLOBAL MEMORY VIA TEST-TIME REGISTERS

## ABSTRACT

A well-known challenge in large attention-based architectures, *e.g.*, Vision Transformers (ViTs) and large language models (LLMs), is the emergence of *attention sinks*, where a small subset of tokens disproportionately attracts attention and ultimately degrades downstream performance. Prior work typically casts these high-norm tokens in ViTs as "computational scratchpads" and analogously characterizes sinks in LLMs as pressure valves that absorb surplus attention with little semantic content. This work revisits that view from the perspective of vision models. It is shown that, in large-scale vision architectures, *register* features in fact encode global, task-relevant information and can act as a plug-and-play global memory. Building on this insight, a training-free framework is proposed that leverages test-time register tokens as global priors to enhance both dense prediction and generative tasks, while mitigating adverse sink effects. In contrast to heuristic placement rules, a theoretically grounded token–neuron interpolation rule is introduced for robust, model-agnostic insertion. Experiments demonstrate improved generative quality and stronger text–image alignment for one-dimensional token generation with ViTs, reflected by gains in FID, IS, CLIPScore, and SigLIP metrics.

## 1 INTRODUCTION

Attention sinks refer to a small subset of tokens attracting a disproportionate fraction of attention mass, which in turn disrupts useful information flow. Recent studies in LLMs and LMMs (Xiao et al., 2024; Sun et al., 2024; Kang et al., 2025; Wang et al., 2025; Gu et al., 2025) frequently attribute such tokens to pressure–valve behavior that absorbs surplus attention with limited semantic content, often at sequence boundaries or visually irrelevant regions. On the vision side, Vision Transformers (ViTs) (Dosovitskiy et al., 2021) exhibit related artifacts: several works (Darcet et al., 2024; Jiang et al., 2025) report that high-norm outlier tokens emerge and attract the majority of attention, commonly aligning with background areas. While works like (Darcet et al., 2024) mitigate this behavior by adding *register tokens* during training as computational scratchpads, (Jiang et al., 2025) identify a sparse set of register neurons whose large activations cause these outliers and introduce test-time registers that can be injected without training. These views are compatible: the neuron-level mechanism explains the token-level emergence of high-norm outliers, and both link to sink-like behavior also observed in StreamingLLM (Xiao et al., 2024) and Sun et al. (2024)s' work.

As Figure 1 illustrates, this paper revisits the prevailing assumption that register features carry little semantic information. Empirically, test-time registers (Jiang et al., 2025) provide stable improvements on downstream tasks, suggesting that the associated features are not merely noise. Building on this observation, a hypothesis is advanced that register features encode global, task-relevant information that behaves as a plug-and-play global memory. To probe this, frequency-domain analyses are conducted on PCA embeddings of token features across OpenCLIP and DINOv2 using 1000 ImageNet images: test-time `[REG]` tokens concentrate more energy in low-frequency bands than `[CLS]` and patch-mean, and the trend persists after whitening. Together with PCA scatter visualizations, these results indicate that registers primarily store slowly varying global content that is distinct from `[CLS]` (task readout) and local patch details; this makes them particularly suitable as global priors for generative and dense prediction tasks. Figure 2 illustrates the trend for both model families. Motivated by this perspective, a training-free framework is proposed that operationalizes registers as plug-and-play global memory at inference. The procedure first identifies intervention locations at the *layer/module* level via a forward-only alignment score, then uses the *register-neuron*

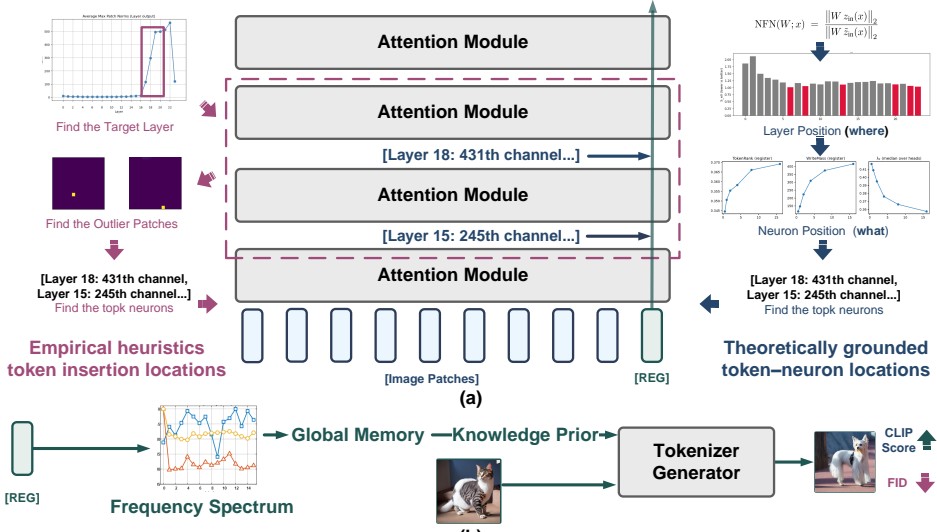

Figure 1: **Overview of the proposed approach.** (a) Unlike prior training-free register insertion methods that rely on empirical heuristics (Jiang et al., 2025) and lack generalization across architectures, our method introduces a theoretically grounded token–neuron interpolation rule for adaptable insertion. (b) Through detailed observation and analysis, we demonstrate that register tokens serve as a global memory and, in particular, significantly enhance token-level generative capabilities.

view to extract and transfer large activations into test-time registers. Finally, a value-aware global-importance analysis bridges token- and neuron-level effects, providing a principled rule for where and how to insert registers. The main contributions can be summarized as follows: **(1)** We revisit register features and show that they concentrate low-frequency bands, global information that is distinct from `[CLS]` and local patches supporting registers as a compact global memory for generation and dense prediction; **(2)** We introduce a practical, reproducible test-time pipeline that uses a theoretically grounded token–neuron interpolation rule for model-agnostic insertion, avoiding empirical heuristics for selecting insertion locations; **(3)** We use test-time registers as plug-and-play global memory (a knowledge prior) to enhance generation, delivering improvements without modifying pretrained weights.

## 2 RELATED WORK

**Attention sink in large attention-based model.** We discuss attention sinks from two complementary perspectives, the token level and the neuron/channel level. In large language models, Xiao et al. (2024) show at the token level that early tokens systematically attract attention and that preserving their KV states stabilizes long-context decoding; they also show that inserting a dedicated sink token works without fine-tuning. Sun et al. (2024) then examine the neuron/channel level, identifying rare input-invariant spikes that behave like implicit biases and concentrate attention largely independent of semantics. Gu et al. (2025) further probe the token-level mechanism, tying sinks to pretraining dynamics and softmax normalization, and observe that non-normalized attention suppresses the effect in sub-1B models. In multimodal decoders, Kang et al. (2025) report the presence of irrelevant visual tokens that absorb attention and demonstrate that redistributing surplus attention (VAR) improves performance. On the vision side, Darcet et al. (2024) document high-norm outlier tokens in low-information regions and show that adding learned register tokens during training suppresses artifacts and smooths features, whereas Jiang et al. (2025) trace these outliers to a sparse set of register neurons and demonstrate that shifting their activations into an extra test-time token reproduces the benefits without training. It is worth mentioning that most prior studies frame sinks/registers as non-semantic or bias-like reservoirs that stabilize attention rather than carriers of task content (Xiao et al., 2024; Sun et al., 2024; Gu et al., 2025; Kang et al., 2025; Darcet et al., 2024).

**Interpretations of transformer attention.** Recent work revisits how to read and operationalize attention beyond raw weights. Yang et al. (2025) propose a training-free sparse-attention rule for

long-form reasoning that aggregates per-head local selections together with a short recency window into a single global token ranking used by subsequent layers; the method preserves or improves accuracy while attending to roughly half as many tokens and yields about $1.1\times$ decoding speed-ups. Hayou et al. (2025) study interpretation at the module level: using Normalized Feature Norms (NFN) computed with forward passes only, they score module–data alignment and place LoRA where alignment is lowest, consistently matching or outperforming common attention-only or MLP-only placement heuristics across supervised finetuning and reinforcement learning for reasoning. Orthogonally, Erel et al. (2025) models each attention matrix as a discrete-time Markov chain; the resulting steady-state TokenRank captures indirect (multi-hop) influence and provides a robust notion of global token importance, improving zero-shot segmentation and unconditional image generation compared with first-order maps. We build on these insights to structure our pipeline: NFN from PLoP guides **where** to intervene (layers/modules), while TokenRank from the Markov view quantifies **which** tokens matter globally; the two meet in a Value×TokenRank bridge that links module-level interventions to token-level effects. This design makes our training-free test-time registers both targeted and interpretable.

**Tokenization for Generative Vision.** Recent work rethinks where generative semantics should live, inside the tokenizer, in a highly compressed latent space, or in an explicit global token. Zha et al. (2025) propose TexTok, which conditions image tokenization on text, allowing visual tokens to focus on fine details while language provides high-level semantics. By simply swapping the tokenizer into a DiT, TexTok achieves strong FID with good inference speedups at only 32 tokens. Lu et al. (2025) introduce a unified visual tokenizer (AToken) that encodes images, videos, and 3D assets into a shared 4D latent space and supports both continuous and discrete tokens, enabling competitive reconstruction and downstream generation across modalities. Orthogonally, Wang et al. (2025) show that vision features for VLMs concentrate energy in low frequencies; a parameter-free DCT/FFT low-pass compresses vision tokens, reducing FLOPs and boosting generation speed with minimal accuracy loss. Pushing compression further, Beyer et al. (2025) demonstrate that a 1D tokenizer (32 discrete tokens) already enables image generation and editing via test-time token optimization with plug-and-play losses (*e.g.*, CLIP or reconstruction), requiring no generative training. Finally, Wu et al. (2025) entangle denoising latents with a single high-level class token throughout diffusion, yielding faster convergence and improved FID while producing coherent image–class pairs from noise. We build on these insights by treating test-time registers as a plug-and-play global prior: compatible with frequency-domain compression (Fourier-VLM), alignable with 1D token spaces for token-optimization (HCT), and complementary to language-conditioned tokenization and class-token entanglement (TexTok/REG), all without retraining the generator.

## 3 OBSERVATION

We turn *attention sinks* – tokens that attract a disproportionate share of attention mass across layers – into a *few-token global memory* for generation, without any retraining. Prior work shows that (i) ViTs exhibit high-norm outlier tokens that can be stabilized by adding *register tokens* (Darcet et al., 2024); (ii) a sparse set of *register neurons* produces these outliers, which enables injecting registers at test time (Jiang et al., 2025); and (iii) LMMs and LRMs universally exhibit attention sinks that act like key biases and help decode long contexts when preserved (Xiao et al., 2024; Gu et al., 2025; Kang et al., 2025). While these studies often frame such tokens as computational scratchpads or pressure valves that mitigate over-mixing with limited semantic content, (Jiang et al., 2025) report that test-time register features achieve competitive linear-probe accuracy on ImageNet (Table 1 in original paper) relative to the [CLS] token. This observation raises two questions: "*What do registers store?*" and "*How do registers differ from other tokens?*"

**Notation and preliminaries.** Let $X^\ell \in \mathbb{R}^{N \times d}$ denote the input token embeddings to the attention layer $\ell$, with $N = 1 + R + P$ comprising 1 [CLS], $R$ [REG], and $P$ [PATCH] tokens. The Attention module can be formulated as follows:

$$Q^\ell = X^\ell W_Q^\ell,\ K^\ell = X^\ell W_K^\ell,\ V^\ell = X^\ell W_V^\ell, \tag{1}$$

$$A^\ell = \mathrm{softmax}\left(\frac{Q^\ell K^{\ell\top}}{\sqrt{d_k}}\right),\quad \mathrm{AttnOut} = A^\ell V^\ell, \tag{2}$$

where $W_Q^\ell, W_K^\ell, W_V^\ell$ are parameter matrices and $A^\ell$ is the attention matrix for a single attention head. We call a token an *attention sink* if it receives persistently large incoming attention mass

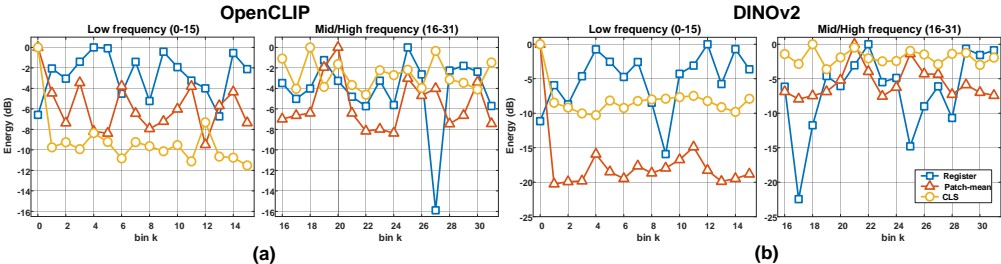

Figure 2: **PCA of token embeddings across layers.** The 1-D FFT spectrum of PCA-projected features is shown for three types of tokens: Register, Patch-mean, and CLS. Notably, the CLS token exhibits higher-frequency components, consistent with its role in aggregating high-level semantic information. In contrast, register tokens display stronger low-frequency components compared to the other two types.

across layers. In vision models such as ViTs, a *register token* is an explicit position that absorbs global or statistical computation (Darcet et al., 2024), and a *register neuron* is a sparse subset of channels whose large activations concentrate on outlier tokens (Jiang et al., 2025). In LLMs and LRMs, initial-token sinks act like key-bias reservoirs (Xiao et al., 2024; Kang et al., 2025; Sun et al., 2024; Beyer et al., 2025; Yang et al., 2025). This paper focuses on large vision models.

### 3.1 WHAT REGISTERS STORE?

**Prior evidence.** ViT studies report high-norm outlier tokens that arise in low-information regions and act as internal workspaces; adding register tokens absorbs these outliers and yields smoother dense features and attention maps (Darcet et al., 2024). A training-free variant identifies sparse register neurons whose activations drive high norms and shows that test-time registers (extra tokens plus neuron activation) replicate the trained-register effect across OpenCLIP and DINOv2 (Jiang et al., 2025). Recent DINOv3 (Siméoni et al., 2025) includes four learned register tokens and introduces Gram anchoring to stabilize dense features, which indicates that modern ViT backbones treat registers as first-class global workspaces.

**Hypothesis and analysis.** Building on these findings and on test-time register results, we hypothesize that registers primarily carry global, low-frequency scene information such as color tone, illumination, and coarse layout, rather than local high-frequency details. We test this hypothesis with a signal-level analysis. Using 1000 ImageNet images, we compute one-dimensional FFT spectra of token features for OpenCLIP and DINOv2, and we compare test-time [REG], the patch mean, and [CLS]. Across models, [REG] concentrates more energy in low-frequency bins (0–15), whereas [CLS] and patch-mean allocate relatively more energy to higher bins (16–31). The trend persists after PCA whitening; see Figure 2a for OpenCLIP and Figure 2b for DINOv2. Together with the PCA scatter, these results support the view that registers serve as compact carriers of global, slowly varying content that is distinct from [CLS] (task readout) and local patch details. Such features naturally serve as a global memory that benefits generation tasks, whereas [CLS] remains strong for discriminative tasks such as classification and retrieval.

## 4 METHOD

Since register features serve as a global memory, and (Jiang et al., 2025) provide a test-time mechanism that reproduces trained-register effects without training, we treat registers as few-token global controls for training-free generation, including style, lighting, and layout. We plug test-time registers and neurons into a token-optimization pipeline, which aligns with recent results showing that very few tokens suffice for inpainting and text-guided editing via test-time optimization. We first need to locate the register-neuron subspace.

### 4.1 LOCATING THE REGISTER SUBSPACE: BRIDGING TOKENS AND NEURONS

The original test-time register procedure (Jiang et al., 2025) proceeds as follows. It identifies a layer $L$ where token-norm or attention-mass statistics rise sharply, selects the top-$n$ high-norm patch

locations $\text{top-}n(\texttt{PATCH})$ in a small window of layers around $L$, and then ranks channels by average activation over these outlier locations across layers preceding $L$ to obtain the top-$k$ register neurons. While effective, this heuristic depends on architecture-specific hyperparameters.

**NFN-based layer/module selection.** Inspired by (Hayou et al., 2025), we first locate where to intervene by measuring *Normalized Feature Norms* (NFN) across layers and submodules. For a module $W$ (*e.g.*, a per-head $Q, K, V$ projection, attention output, or MLP up/down) with input feature $z_{\text{in}}(x)$ on an image $x$, define

$$\text{NFN}(W; x) \;=\; \frac{\big\| W\, z_{\text{in}}(x) \big\|_2}{\big\| W\, \tilde{z}_{\text{in}}(x) \big\|_2}, \tag{3}$$

where $\tilde{z}_{\text{in}}(x)$ is a random vector of the same dimension and norm as $z_{\text{in}}(x)$ with i.i.d. zero-mean Gaussian coordinates. We aggregate over a dataset $\mathcal{D}$ and heads to obtain a layer- and module-level score

$$\text{NFN}_{\ell,m} \;=\; \mathbb{E}_{x \sim \mathcal{D}} \Big[ \text{median}_h\, \text{NFN}(W^{(\ell,m,h)}; x) \Big], \tag{4}$$

with $m \in \{\texttt{Q}, \texttt{K}, \texttt{V}, \texttt{AttnOut}, \texttt{MLP}_\uparrow, \texttt{MLP}_\downarrow\}$. We then condense per-layer evidence by

$$S_\ell \;=\; \min_m\, \text{NFN}_{\ell,m}. \tag{5}$$

Intuitively, $S_\ell \approx 1$ indicates module responses comparable to a matched-norm random baseline, whereas *low $S_\ell$* highlights modules whose responses are disproportionately shaped by a small subset of input directions, a signature we consistently find near sink/register effects. We select a small set of candidate layers $\mathcal{L}_{\text{cand}}$ as the $\kappa$ layers with the lowest $S_\ell$ (typically $\kappa \in [3, 5]$), optionally enforcing a minimum separation to avoid redundant neighbors.

**Token-level localization in candidate layers.** For each $\ell \in \mathcal{L}_{\text{cand}}$ we compute the per-token $\ell_2$-norm map over the last $w$ layers around $\ell$ (*e.g.*, $w = 3$) and automatically detect outlier positions by a robust threshold (median+IQR). We keep the top-$n$ spatial indices $\Omega_\ell = \text{top-}n(\texttt{PATCH})$ that consistently appear as high-norm across the window.

**Neuron-level extraction and the register subspace.** Given $\Omega_\ell$, we rank channels by their mean absolute activations on these outlier tokens, accumulated over a short backward window of $\Delta$ preceding layers (*e.g.*, $\Delta \in \{1, 2, 3\}$):

$$r_d \;=\; \frac{1}{\Delta\, |\Omega_\ell|} \sum_{j=1}^{\Delta} \sum_{p \in \Omega_\ell} \big| a_{p,d}^{(\ell-j)} \big|, \tag{6}$$

where $a_{p,d}^{(\ell-j)}$ denotes the activation of channel $d$ at token $p$ and layer $\ell-j$. We take the top-$k$ channels $\mathcal{R}_\ell = \text{top-}k(\{r_d\})$ as *register neurons* and define the *register subspace* as $\text{span}\{e_d : d \in \mathcal{R}_\ell\}$. This yields a scale-invariant, model-agnostic estimate of where and along which axes sink/register energy concentrates.

**From subspace to test-time registers.** With $\mathcal{R}_\ell$ fixed, we instantiate *test-time registers* by adding one (or a small number) of auxiliary tokens and *transferring* the large activations on $\mathcal{R}_\ell$ from outlier patch tokens to these auxiliary tokens before the residual update. Concretely, let $t_{\text{reg}}$ denote the new token; we modify its value channels on $\mathcal{R}_\ell$ by a scaled copy of the outlier mean while zeroing those channels at the original outlier positions. This preserves task-relevant global statistics in a few tokens and suppresses artifact-causing outliers in patch tokens, without any training. The resulting register tokens will be used downstream as plug-and-play global controls.

**Defaults and practicality.** Unless otherwise noted, we use $\kappa = 5$, $n \in [3, 8]$ (proportional to feature map size), $k \in \{32, 64\}$, $w = 3$, and $\Delta = 2$. We compute NFN on a 1000 ImageNet-val subset. All weights remain frozen. This NFN $\rightarrow$ token $\rightarrow$ neuron pipeline makes the subsequent analysis and interventions reproducible across ViT families and aligns with the training-free nature of test-time registers.

## 4.2 Token-side importance via Value TokenRank

Only inspecting attention weights can be misleading: a token can attract attention but carry little valuable content. Inspired by (Erel et al., 2025), we therefore quantify a token's *effective write* by

combining attention and value magnitudes, and couple it with a global centrality score derived from attention-as-Markov-chains.

**Write mass (per-token write-in).** For an attention head with attention matrix $A \in \mathbb{R}^{T \times T}$ (row-stochastic) and value vectors $\{V_t\}_{t=1}^T$, we define the write mass of target token $t$ as

$$\text{WriteMass}(t) \;=\; \sum_{i=1}^T A_{i,t} \, \|V_t\|_2^2, \tag{7}$$

which better reflects how much content is actually *written* into $t$ than attention alone.

**TokenRank (global centrality).** Viewing $A$ as a discrete-time Markov chain, we define *TokenRank* $\pi \in \Delta^{T-1}$ as the stationary distribution satisfying $\pi^\top = \pi^\top A$. High TokenRank indicates tokens that are globally central under multi-hop attention flow. In practice, we compute $\pi$ per head and average over heads within a layer (or take the maximum for sensitivity analysis).

**Head mixing rate.** Let $1 = \lambda_1 \geq \lambda_2 \geq \cdots$ be the eigenvalues of $A$. The *second eigenvalue* $\lambda_2$ measures how slowly the head mixes (larger $\lambda_2 \to$ stickier dynamics). Empirically, heads/layers with larger $\lambda_2$ align with the emergence of high-norm outliers; after adding test-time registers, both $\lambda_2$ and the correlation between $\lambda_2$ and high-norm frequency decrease.

**Register absorption test.** Given a register token $t_{\text{reg}}$, we scale its value on the register subspace by $s \in \{0.5, 1, 2, 4, 8, 16\}$ and track $\text{TokenRank}(t_{\text{reg}})$, $\text{WriteMass}(t_{\text{reg}})$, and $\lambda_2$. We observe monotone increases that saturate with $s$, indicating an *absorbing-register* behavior desirable for consolidating global statistics.

### 4.3 A token–neuron interpolation rule for robust insertion

Let $\mathcal{R}_\ell$ be the set of register neurons (channels) extracted at layer $\ell$ (Section 4.2), and let $\mathcal{P}$ be the detected outlier patch tokens at the same layer. We form an auxiliary register token $t_{\text{reg}}$ and *reallocate* large activations along the register subspace to $t_{\text{reg}}$ while preserving first- and second-order statistics.

**Projection and conservation.** Write $P_\mathcal{R}$ for the orthogonal projector onto $\text{span}\{e_d : d \in \mathcal{R}_\ell\}$. Let $\bar{v}_\mathcal{R} = \frac{1}{|\mathcal{P}|} \sum_{p \in \mathcal{P}} P_\mathcal{R} V_p$ be the outlier-average on the subspace. For a scale $s > 0$, we set

$$V_{t_{\text{reg}}} \;\leftarrow\; V_{t_{\text{reg}}} \;+\; s\,\bar{v}_\mathcal{R}, \qquad V_p \;\leftarrow\; V_p \;-\; \alpha\, P_\mathcal{R} V_p, \;\; \forall p \in \mathcal{P}, \tag{8}$$

where $\alpha \in (0, 1]$ controls how much register-subspace energy is removed from outliers. Choosing

$$\alpha \;=\; \min\left\{1, \; \frac{s\,\|\bar{v}_\mathcal{R}\|_2}{\frac{1}{|\mathcal{P}|}\sum_{p \in \mathcal{P}} \|P_\mathcal{R} V_p\|_2}\right\} \tag{9}$$

approximately conserves the mean register energy while shifting it from many patches into a few registers, thus suppressing artifacts without losing global statistics.

**Per-head normalization and stability.** To keep layer-wise scales stable, we optionally normalize by head-level Root Mean Square (RMS) on $\mathcal{R}$ before applying Equation 8, and re-center the non-register subspace:

$$V_t \leftarrow V_t - \beta\,(I - P_\mathcal{R})\,\big(\tfrac{1}{T}\sum_{j=1}^T V_j\big), \quad \beta \in \{0, 1\}. \tag{10}$$

This recenters low-frequency biases in the complementary subspace and further reduces checkerboard-like artifacts.

**Complexity and defaults.** We instantiate $|t_{\text{reg}}| = 1$ token per selected layer (or at most two), use $k \in \{32, 64\}$ channels in $\mathcal{R}_\ell$, and $s \in \{1, 2, 4\}$ by default; the procedure is training-free and adds negligible overhead. The whole algorithm is depicted in the Algorithm 1 in the Appendix.

### 4.4 Plug-and-play registers for generation

We close the loop by treating test-time registers as few-token global knowledge for tokenized decoders (*e.g.*, TiTok/HCT). The idea is to map layer-wise register tokens to entries of a fixed codebook and decode images, enabling training-free global edits.

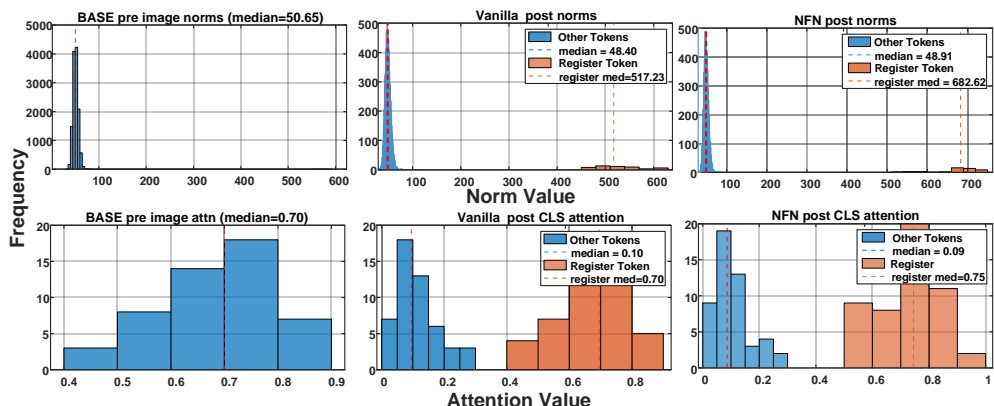

Figure 3: **Distribution before and after register insert on DINOv2.** Adding one register token using NFN-selected layers (top-50 register neurons) yields higher register norm/attention with minimal change to image tokens.

**Register to codebook mapping.** Let $r^{(\ell)} \in \mathbb{R}^D$ be the register token at layer $\ell \in \mathcal{L}_{\text{cand}}$ obtained by Section 4.2 and 4.3. Given a tokenizer codebook $\mathcal{C} = \{c_j \in \mathbb{R}^D\}_{j=1}^M$ with similarity $\sigma(u, v) = \frac{u^\top v}{\|u\|\|v\|}$, we pick the nearest entry

$$j_\ell^\star = \arg \max_{j \in [M]} \sigma\big(r^{(\ell)}, c_j\big), \qquad \tilde{r}^{(\ell)} = c_{j_\ell^\star}. \tag{11}$$

For multiple layers, we fuse by a convex combination

$$\tilde{r} = \sum_{\ell \in \mathcal{L}_{\text{use}}} \alpha_\ell \, \tilde{r}^{(\ell)}, \quad \alpha_\ell \geq 0, \sum_\ell \alpha_\ell = 1, \tag{12}$$

where $\alpha_\ell \propto \text{TokenRank}(t_{\text{reg}}^{(\ell)})$ or uniform.

**Insertion and strength control.** Let $\mathbf{z} = (z_1, \ldots, z_T)$ be the tokenizer token sequence. We insert the fused register vector at a designated global slot (*e.g.*, token #0):

$$z_0 \leftarrow (1 - \gamma) z_0 + \gamma \tilde{r}, \qquad \gamma \in [0, 1], \tag{13}$$

Or replace $K$ early tokens if the decoder expects multiple globals. The scalar $\gamma$ plays the same role as the scale $s$ in Section 4.3 and is swept over $\{0.25, 0.5, 0.75, 1.0\}$.

**Optional test-time token optimization.** To refine semantics without training, we optimize only the inserted token(s) for $S$ steps under a frozen decoder $\mathcal{D}$:

$$\min_{z_0} \mathcal{L}_{\text{CLIP}}\big(\mathcal{D}([z_0, \mathbf{z}_{1:T}]), \text{text}\big) + \lambda \|z_0 - \tilde{r}\|_2^2, \tag{14}$$

with $\lambda \in [0.1, 1]$, step size $\eta \in [1 \times 10^{-3}, 5 \times 10^{-3}]$, $S \in \{50, 100\}$. This keeps $z_0$ close to the register prior while aligning to the prompt.

The whole generation pipeline is depicted in the Algorithm 2 in the Appendix.

## 5 EXPERIMENTS

### 5.1 NFN LAYER LOCALIZATION YIELDS STRONGER ABSORBING BEHAVIOR

**From Outliers to Register Neurons.** We first identify the onset of outliers, $L_{\text{start}}$, using the patch-norm curve, and detect outlier tokens within a small backward window. Within this window, we rank channels to extract the top 50 register neurons, enforcing a per-layer cap and back-filling as needed to ensure exactly 50 are selected. This procedure preserves the original outlier-to-neuron selection pipeline and serves as a shared backbone for both the baseline and our NFN-guided variant. As illustrated in Figure 3, inserting a register token yields the expected absorption effect: both the register value-norm and the CLS→register attention increase substantially, while statistics for image

Table 1: **Norm gap between normal patch and register.** The comparison between the proposed method with different neuron positions and vanilla test-time register on OpenCLIP and DINOv2.

| Method | OpenCLIP (Cherti et al., 2023) | | DINOv2 (Oquab et al., 2024) | |
|---|---|---|---|---|
| | norm gap | norm gap (Attention) | norm gap | norm gap (Attention) |
| Vanilla(50 neuron) (Jiang et al., 2025) | 62.03 | **0.58** | 468.83 | 0.60 |
| NFN-guided (8 neuron) | 61.36 | 0.52 | 485.23 | 0.59 |
| NFN-guided (16 neuron) | 64.42 | 0.55 | 494.63 | 0.63 |
| NFN-guided (50 neuron) | **70.09** | **0.58** | **633.71** | **0.66** |

Table 2: **Effect on outliers and mixing (DINOv2-L/14, ImageNet-val 1k).** Outlier fraction uses the pre-95th percentile threshold fixed for post. $\lambda_2$ is the second eigenvalue (median over heads); we report spectral gap $1 - \lambda_2$ (smaller spectral gap indicates stickier dynamics (i.e., slower mixing)).

| Method | OpenCLIP (Cherti et al., 2023) | | DINOv2 (Oquab et al., 2024) | |
|---|---|---|---|---|
| | $\Delta$ median(outlier) ($\downarrow$) | $\Delta$ gap $= -\Delta\lambda_2$ ($\downarrow$) | $\Delta$ median(outlier) ($\downarrow$) | $\Delta$ gap $= -\Delta\lambda_2$ ($\downarrow$) |
| Vanilla (Jiang et al., 2025) | $-0.013$ | $-0.0024$ | $-0.0215$ | $-0.0036$ |
| Ours | $-0.014$ | $-0.0027$ | $-0.0312$ | $-0.0946$ |

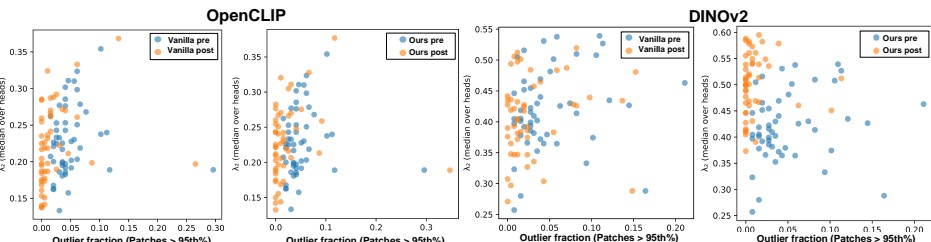

Figure 4: $\lambda_2$**–outlier coupling before/after register insertion.** Each dot is one image (ImageNet-val 1k). $x$–axis: fraction of patch tokens above the *pre*-95th norm threshold (fixed for post). $y$–axis: $\lambda_2$ of the attention Markov chain (median over heads; lower $\lambda_2 \Rightarrow$ larger spectral gap/faster mixing). *Left:* vanilla test-time register yields a mild shift (left/down). *Right:* our method produces a larger leftward shift (fewer outliers) and a clearer downward shift (stronger mixing), consistent with the absorber behavior quantified in Table 2.

tokens remain nearly unchanged. On DINOv2-L/14, for example, the median register value-norm rises from 517.2 to 682.6 (+32%), and the median CLS→register attention increases from 0.70 to 0.75 (+0.05). In contrast, the corresponding median values for image tokens remain steady at approximately 49 and 0.10, respectively. OpenCLIP displays the same trend, with detailed results provided in the Figure 7 in the Appendix. Furthermore, as shown in Table 1, the variant guided by NFN achieves competitive performance compared to the standard test-time register approach, even when using fewer neuron positions, demonstrating the effectiveness of our proposed method.

**Value×TokenRank bridge.** As mentioned in the Section 4.1, attention alone can be misleading since a token may attract probability mass yet write little content. We therefore pair a per-token write measure WriteMass with a global centrality score TokenRank that the stationary distribution of the head-wise attention Markov chain and a mixing diagnostic. Scaling the injected register on its subspace yields monotonic increases in TokenRank and WriteMass and a concurrent drop in $\lambda_2$, indicating controlled absorption into the register rather than patch over-amplification as depicted in Figure 4. Using TokenRank for head gating further strengthens this effect and reduces image-token outliers. As Table 2 shown, quantitatively on DINOv2-L/14 (1k val), vanilla test-time registers reduce median outlier fraction by $-0.0215$ with a companion $\Delta\lambda_2 = -0.0036$, while our NFN-guided+gated variant attains a larger outlier drop of $-0.0312$ and a markedly stronger $\lambda_2$ decrease. The details are described in the Appendix A.2.

## 5.2 PLUG-AND-PLAY REGISTERS FOR GENERATION

**Setting and Metrics.** We adopt ViT-L/14 backbones (DINOv2) and extract register features from layers $L \in \{L, L-1, L-2\}$, selected via NFN. Register vectors are projected into the HCT/TiTok code space using lightweight alignment, either whitening with nearest-neighbor search or a linear

Table 3: **HCT decoding with different global priors.** All methods use the same protocol (VQ-LL-32 codebook, 1000-seed CLIP top-1 association, token optimization), differing only in the injected prior at test time. Lower is better for FID-5k; higher is better for IS/CLIP/SigLIP.

| Method (VQ-LL-32, 1000, CLIP top-1%, token opt.) | FID-5k (↓) | IS (↑) | CLIP (↑) | SigLIP (↑) |
|---|---|---|---|---|
| HCT w/o prior (Beyer et al., 2025) | 21.2 | 281 | 0.40 | 3.5 |
| HCT w/ Random prior | 22.5 | 278 | 0.39 | 3.4 |
| HCT w/ [CLS] prior | 20.5 | 281 | 0.39 | 3.6 |
| HCT w/ test-time register prior (Jiang et al., 2025) | 21.5 | 283 | 0.40 | 3.6 |
| HCT w/ ours | **20.3** | **287** | **0.41** | **3.9** |

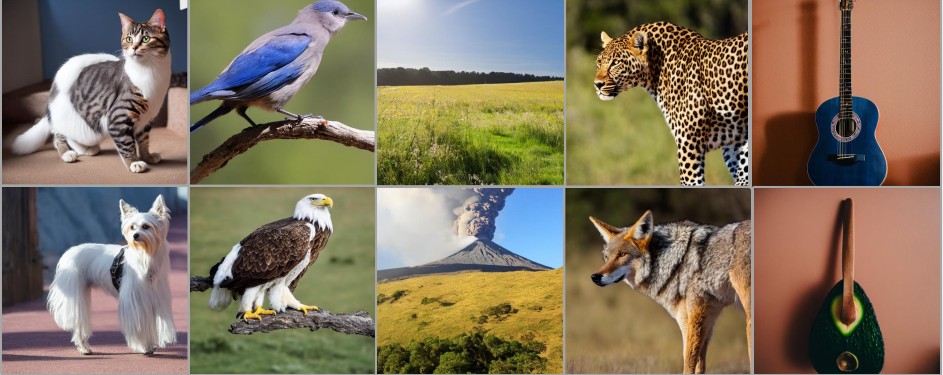

Figure 5: **Qualitative results.** Following HCT evaluation protocol, our proposed method generates reasonable and diverse images from a single input image using the simple prompt `a photo of the [class]`. The top row is input images; the second row is the corresponding generations.

Procrustes method. At decode time, we inject a single global prior in one of two ways: (i) Init-prior: mixing the first code with strength $\gamma \in \{0.25, 0.5, 0.75, 1.0\}$; or (ii) Soft-bias: add a cosine prior to the logits, $l'_c = l_c + \beta \cdot cos(e_c, \hat{u})$, where $\hat{u} = (W_{z_{reg}})/\|W_{z_{reg}}\|$ and $e_c$ is the code embedding. Equivalently, in probability space, $p'(c) \propto p(c) \cdot exp^{(\beta \cdot cos(e_c, \hat{u}))}$. with $\beta \in \{0.5, 1, 2\}$. Optionally, we optimize the first $m$ codes for 20–50 steps using a CLIP loss with a small $\ell_2$ regularization toward the injected prior. All other HCT hyperparameters (VQ-LL-32 codebook, 1000-seed CLIP top-1 association, decoding temperature, CLIP crops/EMA) are held fixed across all methods. In practice, we select $\beta$ on a held-out validation split and report test metrics at the selected $\beta = 1$, adopting the soft-bias.

**Baselines.** We compare against: No prior, Random prior (random vector matching the register's norm), [CLS] prior, and Test-time register prior (Jiang et al., 2025) (TTR). The proposed method utilizes NFN-guided layer selection and TokenRank head gating to form a register-based prior.

**Results.** Table 3 summarizes quantitative results for the token optimization setting. As discussed in Section 4.2, stronger absorber behavior (*i.e.*, reduced outlier tokens and spectral gap $1-\lambda_2$) yields smoother global style and fewer checkerboard artifacts in the generated images. The proposed method consistently outperforms Random and [CLS] priors, and matches or surpasses the training-free baseline, all without any model retraining. Figure 5 presents qualitative examples, illustrating the variability and visual quality enabled by register-based priors.

# 6 CONCLUSION

This work presents a plug-and-play global memory framework that injects a test-time register as a global prior, grounded in empirical observations and a simple theory. The analysis shows that, in large-scale vision architectures, register features encode global, task-relevant information and can be repurposed as reusable memory. A theoretically motivated token–neuron interpolation rule enables robust, model-agnostic insertion without tuning architecture-specific hyperparameters. Leveraging the test-time register as prior knowledge improves 1D token generation, yielding higher visual quality and stronger text–image alignment. These results reposition registers beyond mere computational scratchpads and highlight their potential as general-purpose priors for generative vision tasks.

**Ethics statement.** We confirm that this research adheres to the ICLR Code of Ethics. Our work is built upon publicly available datasets and pretrained models. We have evaluated our methodology and foresee no direct negative societal impacts or ethical concerns arising from this research.

**Reproducibility statement.** To ensure the reproducibility of our research, the source code required to replicate the experiments presented in this paper will be made publicly available upon publication.

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

# A APPENDIX

## A.1 MORE ANALYSIS RESULTS OF MAIN COMPONENTS

**NFN-based layer/module localization** We compute NFN scores per module and aggregate by $S_\ell = \min_m \text{NFN}_{\ell,m}$ over a 1000 ImageNet-val subset. The lowest-$S_\ell$ layers are selected as $\kappa=5$ candidates. We report the distribution of $S_\ell$, cross-backbone consistency, and visualize the a layer×module heatmap of DINOv2 in Figure 6.

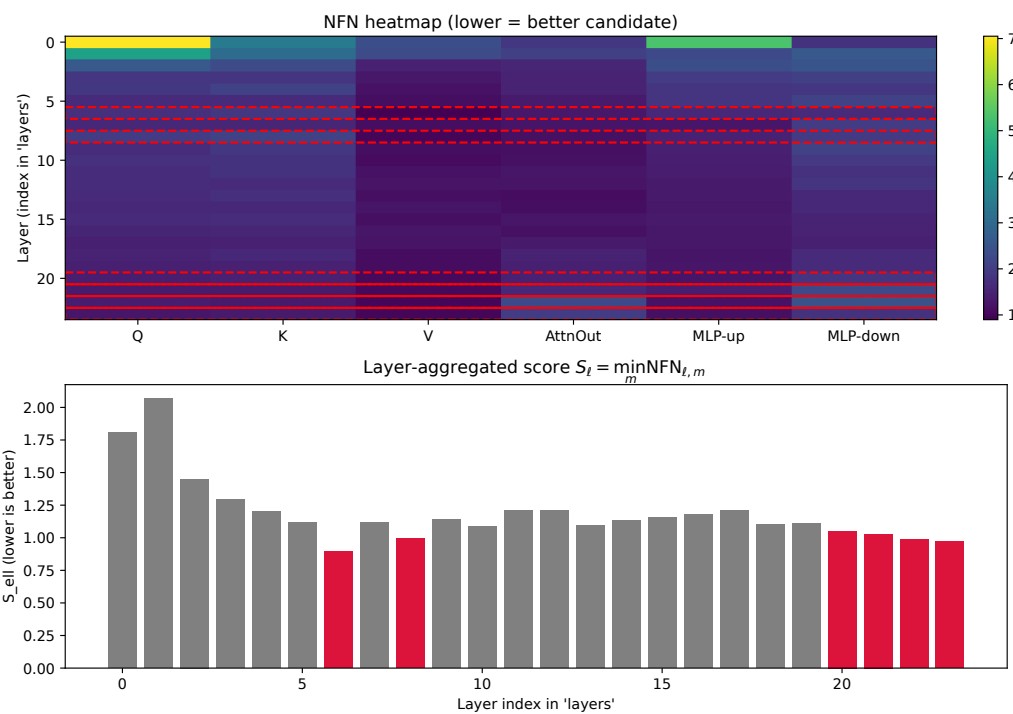

Figure 6: Heatmap visualization and candidate layer positions identified by NFN.

**NFN layer localization yields stronger absorbing behavior** OpenCLIP displays the same trend as DINOv2, as Figure 7 illustrates, the median register value-norm rises from 84.91 to 92.95 (+10.5%), and the median CLS → register attention and the corresponding median values for image tokens are nearly the same.

## A.2 DETAILS FOR THE VALUE TO TOKENRANK BRIDGE

We couple a per-token write measure with a global centrality score to avoid interpreting raw attention. For a head with attention $A \in \mathbb{R}^{T \times T}$ (row–stochastic) and values $\{v_t\}$, we define the *write mass* of token $t$ as

$$\text{WriteMass}(t) = \sum_{i=1}^{T} A_{i,t} \|v_t\|_2^2,$$

which better reflects how much content is written *into* $t$ than attention alone. Treating $A$ as a Markov chain, *TokenRank* is the stationary distribution $\pi^\top = \pi^\top A$ (computed per head and then median–aggregated within a layer). We quantify head mixing by the second eigenvalue $\lambda_2$ of $A$ (smaller $\lambda_2 \Leftrightarrow$ larger spectral gap, faster mixing). In practice, we use post–softmax $A$, power iteration for $\pi$, and report per–layer medians across heads for $\pi(\text{REG})$, $\text{WriteMass}(\text{REG})$, and $\lambda_2$.

To assay absorption, we scale the register coordinates on their subspace by $s \in \{0.5, 1, 2, 4, 8, 16\}$ and plot three curves: TokenRank(REG), WriteMass(REG), and the median $\lambda_2$. Monotone increases in the first two with a concurrent decrease in $\lambda_2$ indicate that the added token acts as a controlled

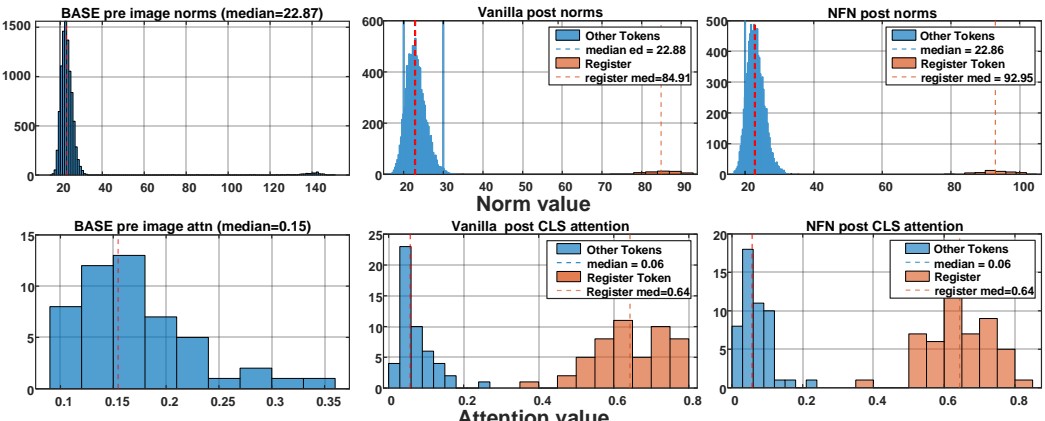

Figure 7: **Distribution before and after register insert on OpenCLIP.** Adding one register token using NFN-selected layers (top-50 register neurons) yields higher register norm/attention with minimal change to image tokens.

absorber rather than amplifying patch outliers. We also use TokenRank to gate heads: at a candidate layer, we keep the top-$h$ heads by $\pi(\texttt{REG})$ (typically $h = 1$–$3$) and apply the test-time register only on those heads. This strengthens absorption and reduces image-token outliers, yielding larger shifts in the $\lambda_2$–outlier joint plot than the vanilla baseline. All experiments share the same implementation details as in the main text.

## A.3 ALGORITHM

---

**Algorithm 1** NFN→DNR→TokenRank: training-free registers

---

**Require:** ViT, dataset $\mathcal{D}$, image batch $\mathcal{B}, \kappa, n, k, w, \Delta, s$
  1: Compute $\text{NFN}_{\ell,m}$ over $\mathcal{D}$; pick $\mathcal{L}_{\text{cand}} = \text{top-}\kappa$ layers with smallest $S_\ell = \min_m \text{NFN}_{\ell,m}$
  2: **for** $\ell \in \mathcal{L}_{\text{cand}}$ **do**
  3:     Detect outlier tokens $\mathcal{P}$ via $\ell_2$-norm maps across a window of $w$ layers
  4:     Rank channels by mean $|a_{p,d}|$ over $p \in \mathcal{P}$ and previous $\Delta$ layers; take $\mathcal{R}_\ell = \text{top-}k$
  5:     Form $t_{\text{reg}}$ and update values by Equation 8 with scale $s$
  6: **end for**
  7: For each head, compute WriteMass, $\pi$ (TokenRank), and $\lambda_2$; log curves vs. $s$
  8: **return** Modified forward pass with test-time registers

---

**Algorithm 2** Register-guided decoding (training-free)

---

**Require:** Register tokens $\{r^{(\ell)}\}$, codebook $\mathcal{C}$, base tokens $\mathbf{z}$, weights $\{\alpha_\ell\}$, strength $\gamma$
  1: **for** each $\ell \in \mathcal{L}_{\text{use}}$ **do**
  2:     $j_\ell^\star \leftarrow \arg\max_j \sigma(r^{(\ell)}, c_j), \quad \tilde{r}^{(\ell)} \leftarrow c_{j_\ell^\star}$
  3: **end for**
  4: $\tilde{r} \leftarrow \sum_\ell \alpha_\ell \tilde{r}^{(\ell)}$
  5: $z_0 \leftarrow (1 - \gamma)z_0 + \gamma \tilde{r}$                          ▷ insert as global prior
  6: **return** decoded image $\mathcal{D}([z_0, \mathbf{z}_{1:T}])$

---

## A.4 LIMITATION

While the primary focus of this work is on large vision models, it is important to note that the phenomenon of attention sinks also arises in large language models (LLMs), large multimodal models (LMMs), and large retrieval models (LRMs). Although the underlying mechanism is conceptually similar across these architectures, our proposed approach has not yet been empirically validated on non-vision models. Furthermore, while our method demonstrates generative capabilities

for tokenizer-based generative models, there exist more sophisticated generative architectures with attention modules (*e.g.*, DiT). Investigating the role of attention sinks in such models remains an open avenue for future research.

### A.5 THE USE OF LARGE LANGUAGE MODELS (LLMS)

We used a large language model (*i.e.*, ChatGPT and Gemini) only for language polishing, including grammar correction, phrasing/clarity improvements, and typographical edits. All outputs were manually reviewed. No LLM contributed to scientific content, and LLMs are not eligible for authorship. We did not share any confidential, reviewer-only, or identifying information with an LLM.

