# OpenReview forum: "Plug-and-Play Global Memory via Test-Time Registers"
_ICLR.cc/2026/Conference — ICLR 2026 Conference Withdrawn Submission_

### Official Review · Reviewer_zmKk · 2025-10-25

**Soundness:** 2
**Presentation:** 2
**Contribution:** 2
**Rating:** 4
**Confidence:** 3

**Summary:**

This paper proposes Plug-and-Play Global Memory via Test-Time Registers, a training-free framework that utilizes Vision Transformer register tokens as global priors during inference. It argues that register features encode global, low-frequency, task-relevant information rather than acting as noise or “attention sinks.” The method introduces a theoretically motivated token–neuron interpolation rule to insert registers robustly across architectures, yielding measurable improvements in generative quality and text-image alignment on benchmarks such as OpenCLIP and DINOv2.

**Strengths:**

1. The paper presents a perspective on register features, demonstrating that they serve as a compact global memory by concentrating low-frequency, global information distinct from [CLS] and local patch representations.
2. Its plug-and-play nature enables performance improvements in generation without altering pretrained weights.

**Weaknesses:**

1. The paper lacks ablation experiments to clarify the individual contribution of components like the NFN layer selection, TokenRank gating, and the interpolation rule.
2. The method introduces many hyperparameters (e.g., k, n, w, s, α) without clear guidance or justification for their choices.
3. Compared to the strong baseline Vision Transformers Don’t Need Trained Registers (Jiang et al., 2025), the performance improvements are minor (FID 21.5->20.3, IS 283->287), raising questions about novelty and impact.
4. There is limited analysis of sensitivity or generalization, making it unclear how robust the method remains across architectures or tasks.

**Questions:**

See weakness.

---

### Official Review · Reviewer_zV3Q · 2025-10-28

**Soundness:** 2
**Presentation:** 2
**Contribution:** 2
**Rating:** 4
**Confidence:** 2

**Summary:**

This paper proposes that attention sinks/register tokens encode global information in Vision Transformers and can be used to control generation. The authors use NFN (Normalized Feature Norm) to identify outlier neurons and transfer outlier activations to inserted register tokens. They validate the approach on OpenCLIP and DINOv2, and demonstrate generation results using HCT decoding, showing modest improvements over prior methods.

**Strengths:**

1.  Statistical analysis of information across different tokens
2. The proposed register tokens effectively absorb outliers and attention sinks

**Weaknesses:**

1. The claim about "encoding global information" has no direct relationship with the subsequent method.
2. Although the authors criticize prior work for being architecture-specific with many hyperparameters, the proposed method appears more complex and requires even more hyperparameters.
3. Despite claiming that registers can serve as "priors" or "encode global, task-relevant information and can be repurposed as reusable memory," no direct experiments substantiate these claims.
4. Unclear analysis:
  - Which specific vectors are used for FFT to obtain frequency components?
  - Why do modules with low NFN indicate outlier neurons?

**Questions:**

##  1. Section 3.1 FFT analysis details:

1.1 Line 195 mentions applying FFT to "token features." Given that channels are exchangeable, is it reasonable to perform FFT across the channel dimension?

1.2 Line 171 mentions using "PCA-projected features." Which implementation is actually used—FFT on raw features or on PCA-projected features?

1.3 If FFT is applied after PCA projection, does this have statistical meaning, and how should the results be interpreted?

## 2. Why does low NFN indicate register tokens?

The paper selects layers with the lowest NFN scores to identify register/sink phenomena. However, wouldn't NFN >> 1 better indicate that most neurons are activated by specific input directions?

## 3.  How are register tokens used as global/task-relevant information in experiments?

The paper claims register tokens encode "global, task-relevant information," but how is this demonstrated experimentally?
 In Section 5.2, are the register features extracted from the same image being generated, or from different images? If from the same image, how is this a "prior" rather than self-conditioning?

---

### Official Review · Reviewer_3pSt · 2025-10-31

**Soundness:** 2
**Presentation:** 1
**Contribution:** 2
**Rating:** 2
**Confidence:** 4

**Summary:**

This work proposes a method to rework, add, or retrofit test time registry tokens [1] in pre-trained foundation models to act as latent context tokens, similar to existing latent token approaches such as TiTok [2]. It is a little unclear what the authors intend at the outset of the manuscript, since they outline a study of existing registry tokens, yet seemingly wants to construct additional registry tokens at test time via interpolation, which this reviewer reads as the main contribution of the work. The final explicitly claimed contribution is made regarding generative tasks.

The aforementioned study (the first contribution) is motivated with arguments from statistical signal processing using 1000 images from ImageNet, which the authors thereby claim demonstrates how registry tokens have “more low frequency energy”.  The authors argue that this shows evidence for “global memory”.  The motivation and central methodology of this study comes across as unclear, potentially flawed, and seemingly unrelated to the central contribution of the paper.

The authors then present a method based on existing work (NFN and TokenRank) providing a method for identifying where and how to insert test-time register tokens as a kind of plug-and-play global memory without retraining. This is performed via an interpolation method, which is what this reviewer reads as the central contribution. The third contribution is an evaluation of generative approaches with HCT [3] (this abbreviation is not connected to a citation in the related work, and is inferred form the readers perspective).

Overall, the paper comes across as unclear or underwritten, with experiments and studies that seem only tangentially related to the key contributions.

[1] [Darcet et al. 2024 - Vision Transformers need Registers](https://arxiv.org/abs/2309.16588)

[2] [Yu et al. 2024 - An Image is Worth 32 Tokens for Reconstruction and Generation](https://arxiv.org/abs/2406.07550)

[3] [Beyer et al. 2025 - Highly Compressed Tokenizer Can Generate Without Training](https://arxiv.org/abs/2506.08257)

**Strengths:**

1. The most plausible source of novelty here is the projection and conservation interpolation rule (Sec. 4.3) (Eqs. 8–10), which (if this reviewer understands the work correctly) proposes to shift register-subspace energy from outlier patches into a test-time register, while stabilizing scales. By contrast, the NFN layer selection and TokenRank gating are repurposed diagnostics from prior work.
2. In a more generous reading, the authors seem to attempt to connect token- and neuron-level diagnostics into a single pipeline. That kind of idea is conceptually interesting as an effort toward unifying perspectives across the two dimensions.

If the “training-free” fitting of inference time latents-from-registers fully live up to the authors stated claims, then this is a clear strength. However, the presentation in the current manuscript is currently too unclear for this reviewer to agree that this is convincingly established as a result.

**Weaknesses:**

1. The motivation for the statistical image processing study is not clearly related to the contributions of the work. The authors seek to establish properties with train-time registry tokens, however, since the proposed method is exclusively based on new test-time registry tokens, this link is not clear to this reviewer. What exactly does this study tell us about the new “plug-and-play” memory tokens the authors propose?
2. The theoretical basis and methodology used in the motivating study is dubious. The authors are mixing concepts in interpreting PCA as a generalised Fourier transform (a valid equivalence in specific circumstances) with harmonic analysis of the PCA over 1000 sampled image tokens. This latter view is problematic, since taking the FT of the PCA over tokens imposes an order on the dimensions of the embedding which is not inherently meaningful, and does not correspond to spatial frequency in a manner which the authors argue. Either this reviewer is missing something, or this is a flawed method for showing some vague “global” property in the registry tokens.
3. There seems to be a bit of circular reasoning with regard to the initial study; the authors assume registers encode “global context,” then define “low-frequency” as global, and unsurprisingly find that registers are “low-frequency.” It’s a form of self-confirming analysis.
4. The sample size for the study is arbitrarily limited to 1000 images; which says very little about ImageNet as a whole. At a minimum, the study could easily be extended to 50k to 100k samples without much computational overhead, matching the validation fold as a bare minimum. A PCA using less than 0.1% of samples is prone to noise and spurious correlations. Also, this reviewer cannot seem to find a reference to HOW these were sampled; making the confusion even worse. Are all classes represented? Is it uniformly sampled? The reader is left guessing, which makes the study difficult to accept as valid.
5. The explanations behind the key method does not provide a very clear picture of the main contribution. Section 4 makes an attempt to mathematical exposition the key components in the method, but misses the mark. Figure 1 did not assist this reviewer in understanding the method either.
6. There is a worrying ambiguity between the claimed contributions in the paper and existing work. The result is confusing; the reader can’t make heads or tails of what the central contributions are, and what has already been proposed by other researchers.
    - A central method HCT is not properly introduced. The authors seem to treat it as shorthand for “Highly Compressed Tokenizer” [3], but it’s never defined. Likewise, “Fourier-VLM” and “TexTok/REG” are mentioned only as narrative hooks, with no bibliographic entries or in-text citations  despite the reference list including papers with those titles, the connections aren’t established in-text.
    - The central contribution of the current paper is arguably the interpolation rule (projection and conservation) in addition to the pre-head normalisation and stability (Sec 4.3). This comes after establishing NFN and TokenRank as central components, which makes it difficult to parse where the contribution of the authors actually lie. This can likely be fixed easily, but in the context of ambiguous citations and unclear contributions, it comes across as unfinished.
7. This reviewer has a central concern with the evaluation of the work, as it seems that the evaluation lacks coherence with the stated motivation. The authors measure token diagnostic metrics, and generative performance. The diagnostic metrics (token norms, outlier ratios) are internal to ViTs and confirm only that injected registers alter attention statistics. The subsequent generative tests, performed through HCT/VQ-LL-32 decoding pipeline, are disconnected from this analysis and only marginally support the “global prior” claim. No evidence is provided that the proposed interpolation rule improves standard vision tasks (classification, retrieval, or segmentation), nor is the choice of HCT justified or reproducible. As a result, the evaluation does not convincingly demonstrate the practical utility of the method at the level that is currently claimed.

**Questions:**

1. What is the intended role of the PCA–FFT analysis in motivating or validating the proposed test-time registers? How does this study on train-time registers inform the design or behaviour of the new “plug-and-play” memory tokens introduced later?
2. The paper risks circularity in equating “low-frequency” with “global” and then concluding that registers are global because they are low-frequency. How do the authors operationalize “global” without invoking the frequency proxy (e.g., explicit global attribute metrics or controls) to avoid this loop?
3. Could the authors clearly state which components of the proposed method are novel? For instance, is the projection-and-conservation interpolation rule in Section 4.3 (Eqs. 8–10) entirely original, or partly adapted from prior work? How do NFN and TokenRank differ from their original formulations?
4. How sensitive are the reported results to the inclusion of each component? Have the authors tested variants with only NFN, only TokenRank, or only the interpolation rule to isolate their respective effects?
5. Can the authors confirm that HCT refers to “Highly Compressed Tokenizer” [3]? Why was this pipeline chosen as the evaluation setting for “global priors,” and what are its implementation details (codebook, optimization steps, β / γ sweeps, seeds)? Why aren’t other baselines evaluated?
6. Are the small reported improvements in Table 3 statistically significant? How many random seeds were used, and what is the variance across runs?
7. In the PCA-to-FFT study, over what variable is the one-dimensional FFT computed, and why is that ordering semantically meaningful? Would whitening not remove the very correlations being interpreted as low-frequency energy?
8. How were the 1000 ImageNet images sampled for that analysis? Do the results hold when the sample size is increased?
9. If register tokens are hypothesized to encode “global” visual attributes, have the authors tested this directly? For example, by quantifying changes in global image properties (illumination, color, layout) when injecting their registers versus a [CLS] or random prior? Can the authors convincingly include such experiments?
10. Have the authors examined whether the proposed method improves any discriminative or dense prediction tasks, such as classification, retrieval, or segmentation, where attention structure is directly relevant? If not, why is this omitted, and why is the evaluation on generative tasks related to the study on registry tokens in foundation models?
11. Several aspects of the implementation and evaluation remain unclear. Could the authors comment on whether they believe the study is reproducible from the information currently provided in the text?

### Suggestions

Separate from these questions, this reviewer would suggest the authors strive to rhetorically and logically connect the contributions to the studies and experimental results of the paper. The authors are also encouraged to demonstrate the feasibility of their method to non-generative tasks, which seem somewhat "bolted on" with regards to the theoretical motivation in the paper. Lastly, restructuring the mathematical exposition by having a small summary in prose before each section would help readability, and clarify the goals as the method is developed throughout the text.

---

### Official Review · Reviewer_bken · 2025-11-01

**Soundness:** 3
**Presentation:** 2
**Contribution:** 3
**Rating:** 4
**Confidence:** 3

**Summary:**

This paper revisits “attention sinks” in large attention-based models and argues that register tokens encode global, low-frequency, task-relevant information that can be used as a plug-and-play global memory at test time. The authors propose a training-free pipeline that (i) selects intervention layers with Normalized Feature Norms, (ii) extracts a register subspace by ranking high-activation channels around outlier patches, (iii) introduces a token–neuron interpolation rule that conserves energy while shifting content from outlier patches into new test-time register tokens, and (iv) quantifies/controls token influence with TokenRankcoupled with a value “write mass” measure. They then use the resulting register vector as a global prior in 1-D token generative decoders, reporting small but consistent gains on FID, IS, CLIPScore, SigLIP and reductions in outlier patches / faster attention mixing.

**Strengths:**

1)	The paper’s core technical novelty is concrete and useful: it introduces an NFN-guided placement plus a token–neuron interpolation rule that bridges token-level sinks and neuron-level activations. The proposed method clearly explains several key decisions including 1) how to locate layers, 2) which channels to move, and 3) how to move them.
2)	The work proposes TokenRank with a value write-mass measure. This value indicates whether the inserted registers truly become global absorbers and whether patch outliers are suppressed.
3)	The framework is training free. The framework adds only a few tokens, and uses forward-pass statistics. This makes it easy to adopt in frozen pipelines.

**Weaknesses:**

1. Your pipeline is close to Jiang et al., 2025 (test-time registers) and Darcet et al., 2024 (trained registers); the distinct elements are NFN-guided placement, token–neuron interpolation, and TokenRank gating. The impacts of the proposed elements are not clearly presented in the paper. To attribute your contribution and quantify its unique impact, could you run ablations to explain the effectiveness of each components?
2. To position your method as the preferable test-time control under matched conditions, could you include these training-free baselines on the same Fourier-VLM/HCT/TiTok (referred in Section 2 in your paper) pipeline and report quality–latency trade-offs?
3. The paper argues generality (LLMs/LMMs/DiT) but evaluates only ViTs and 1D token decoders. Could you provide supported material for this claim?

**Questions:**

Please see the weaknesses.

---

### Note · Authors · 2025-11-12

I have read and agree with the venue's withdrawal policy on behalf of myself and my co-authors.